# Physical Activity and Sedentary Behavior Research in Indonesian Youth: A Scoping Review

**DOI:** 10.3390/ijerph17207665

**Published:** 2020-10-21

**Authors:** Fitria D. Andriyani, Stuart J.H. Biddle, Novita I. Arovah, Katrien De Cocker

**Affiliations:** 1Physically Active Lifestyles Research Group (USQ PALs), Centre for Health Research, University of Southern Queensland, Springfield 4300, Australia; Stuart.Biddle@usq.edu.au (S.J.H.B.); Katrien.DeCocker@usq.edu.au (K.D.C.); 2Department of Sports Education, Faculty of Sports Science, Yogyakarta State University, Yogyakarta 55281, Indonesia; 3Department of Sports Science, Faculty of Sports Science, Yogyakarta State University, Yogyakarta 55281, Indonesia; novita@uny.ac.id

**Keywords:** health, young people, low- and middle-income countries, Indonesia

## Abstract

Background: This study aimed to map physical activity and sedentary behaviour research trends, designs, and topics for Indonesian youth. Methods: This review conforms to the “Preferred Reporting Items for Systematic reviews and Meta-Analyses extension for Scoping Reviews (PRISMA-ScR).” A systematic search on eight platforms was performed in August 2018 and was updated in April 2020. Results: From 10,753 documents screened, 166 met the selection criteria. Over half of the studies were cross-sectional, and the majority utilized self-reported measurements (physical activity: 81.1%, sedentary behavior: 88.5%). More than two-thirds of the studies examined physical activity only (67.5%). The top three subtopics reported were prevalence/measurement, correlates, and outcomes of physical activity (28%, 24.6%, and 17%, respectively). The prevalence of “sufficient” physical activity ranges between 12.2% and 52.3%, while the prevalence of sedentary behavior ≥3 h per day ranges between 24.5% and 33.8%. Conclusions: Future studies need to focus more on intervention and validation, and research needs to be conducted more with nationally representative samples and on youth at the junior high school level. Future studies need to investigate more on psychological, cognitive, affective, social, cultural, and environmental correlates, and in-depth personal views of physical activity and sedentary behavior. More studies using device-based measurements, longitudinal designs, as well as qualitative and mixed-methods approaches are warranted.

## 1. Introduction

More than one-third of the world’s population comprises young people (<20 years old) and in 2017 more than 2.1 billion of them were affected by non-communicable diseases (NCDs), such as cardiovascular diseases, poor mental health, chronic respiratory disorder and diabetes [1]. The high incidence of NCDs among young people has presented a significant public health burden. Youth with NCDs are more likely to face a long-lasting challenge to control or reverse their conditions [1]. Thus, early prevention is the best option.

Insufficient physical activity and high sedentary behavior are among the key drivers of non-communicable diseases (NCDs) in youth [1,2]. These behaviors are often established and reinforced during adolescence and can track over time, this contributing to diseases later in life [3]. Therefore, improving physical activity and reducing sedentary behaviors should be a mainstay in NCD prevention among young people. Unfortunately, these behaviors have often been ignored in low- and middle-income countries (LMICs) [4]. This is shown by the high prevalence of insufficient physical activity and sedentary behavior among youth in LMICs.

It was recommended by the World Health Organization (WHO) in 2010 that young people should accumulate at least 60 min of moderate-to-vigorous-intensity physical activity (MVPA) every day [5] and other national guidelines, such as Australia and Canada, advise young people limit their sedentary time, specifically recreational screen time, to a maximum of 2 h per day [6,7]. The WHO are updating their guidelines in 2020. However, in a study involving 49 LMICs, it was found that less than 30% of adolescents met the physical activity guideline [8]. Moreover, data from 66 LMICs showed that 26.4% of adolescents had a prevalence of sedentary behavior of ≥3 h per day [9]. The global action plan on physical activity 2018–2030 [10] is clear in its message that more needs to be done for LMICs.

It is crucial to pay attention to active and sedentary behaviors in LMICs as more than 80% of the global population lives in these countries and 80% of NCDs are located here [11]. Studies on physical activity and sedentary behavior in these countries are rather few, showing a gap between where research is taking place and the location where public health problems are evident [11]. Therefore, more studies on physical activity and sedentary behavior need to be done in these countries, including Indonesia.

Indonesia is one of the LMICs in the South-East Asia region, with a population of more than 260 million [12,13]. Young people (<20 years old) account for more than 92 million of the total population, which is the fourth largest child population in the world [13,14]. The WHO estimates that the proportion of mortality due to NCDs has increased significantly in Indonesia from 50.7% in 2004 to 71% in 2014 [15]. The development of the economy, and the increasing use of motorized transport and physically less demanding occupations, has caused an increase in the prevalence of physical inactivity and sedentary lifestyles [15]. This is also true for young people, who are typically the most active segment of society.

To guide future research and policy in Indonesia, it is important to know what the current situation is regarding physical activity and sedentary behavior literature in youth in the country. The majority of reviews on physical activity and sedentary behavior include English language studies only, which may exclude studies from LMICs [16], including Indonesia. To our knowledge, there is no study reviewing physical activity and sedentary behavior literature in Indonesian children and adolescents. The unique characteristics of Indonesia, i.e., an archipelago country, which consists of thousand islands with a large youth population, may provide interesting insight into this field of study among LMICs. Therefore, the current paper reports physical activity and sedentary behavior studies in Indonesian youth.

The reason for choosing a scoping review, instead of a conventional systematic review, is because of its suitability with the objective of this study, including its ability to examine the range of available evidence of any method irrespective of its quality [17]. In addition, this type of review is useful to map existing research patterns, and to investigate the implementation of research on a specific field and to find and analyze gaps in existing studies [18,19]. In this systematic scoping review, the purpose was to identify physical activity and sedentary behavior studies on Indonesian youth to map topics and trends for public health. Specifically, trends were assessed for both physical activity and sedentary behavior concerning research topics, research designs used, sample characteristics and measurement methods adopted. Such an analysis aims to identify gaps in the literature and propose recommendations for future research.

## 2. Methods

### 2.1. Literature Search

This scoping review conforms to the “Preferred Reporting Items for Systematic reviews and Meta-Analyses extension for Scoping Reviews (PRISMA-ScR)” [20] (See Appendix A). A literature search was performed in August 2018 and was updated in April 2020 to capture studies in both the Indonesian and English languages. The following platforms were accessed: (1) EBSCOhost Megafile ultimate (Academic Search Ultimate, CINAHL, Education Research Complete, E-Journals, Health Source: Nursing/Academic Edition, MasterFILE Premier, PsycINFO, SPORTDiscus); (2) PubMed; (3) ProQuest dissertations and theses A&I; (4) Web of Science (MEDLINE, Science Citation Index Expanded 1985–present, Social Sciences Citation Index 1985–present, Arts & Humanities Citation Index 1985–present, Conference Proceedings Citation Index-Science 1990–present, Conference Proceedings Citation Index—Social Science & Humanities 1990–present, and Emerging Sources Citation Index 2015–present); (5) Google Scholar; (6) Google; (7) Neliti (Indonesian scientific repository); and (8) Electronic Theses & Dissertations (ETD) Gadjah Mada University. The detailed search strategies can be seen in Appendix A.

### 2.2. Inclusion Criteria

Studies were included in this scoping review if they: (1) targeted Indonesian male and/or female children and adolescents, age 7–18 years old; (2) reported physical activity, physical inactivity, and/or sedentary behavior; (3) were written in the Indonesian and/ or English language, and (4) were published as a journal article, conference proceeding, thesis at Master or Doctoral level in full or abstract form, or report. Any research designs were eligible for inclusion.

Studies were excluded if: (1) they targeted Indonesian populations who live overseas, (2) they focused on sports performance, coaching, and/or physical education, (3) they were published as literature reviews, (4) they did not provide clear information about the age or the school level of the participants, or (5) a full text was not available, except for student theses.

### 2.3. Study Selection and Data Extraction

All references were imported into EndNote X8 software (Clarivate Analytics, Philadelphia, United States). After removing duplicates, the records were screened in three stages—by title, abstract, and full text. Two independent reviewers (FDA and NIA) screened the titles. After that, FDA screened the abstracts. SJHB and KDC screened 18% of the abstracts (*n* = 172) to check the inter-reviewer reliability (agreement: 94%). All discrepancies were resolved through discussion. In the final stage, FDA screened all full texts. Figure 1 illustrates the flow of the searching and screening process.

A data extraction form was created in MS Excel by adapting an existing data extraction form [16]. FDA independently extracted the records. Key data extracted are bibliographic characteristics, study topics, designs, characteristics of study samples, measurements, and study results. The evidence is presented in a descriptive narrative format.

## 3. Results

### 3.1. Bibliographic Characteristics

In total, we screened 10,453 documents with 166 studies meeting the selection criteria (see Figure 1). The selected literature was published between 1998 and 2020. The first study reporting physical activity was an intervention study in 1998 [21]. Meanwhile, the first study reporting sedentary behavior was a case-control study in 2004 [22].

Up to 2011, the number of papers published in the physical activity and sedentary behavior field in Indonesia was relatively low. There was an apparent increase in publications after 2011, mainly on physical activity. This trend continued and reached its peak in 2017 before gradually decreasing in the following years. Figure 2 shows the number of studies up to 2019 (the literature search in 2020 was conducted only until April). There were eight studies published from January to April 2020 [23,24,25,26,27,28,29,30].

Most studies were published as journal articles (80.7%). Other publication types were much fewer (conference proceeding: 10.2%, theses: 5.4%, government documents: 3%, and reports: 0.6%). The full-text and abstract availabilities were 95.8% and 98.2%, respectively. Nearly two-thirds of the full-texts were in the Indonesian language (Bahasa), and half of the abstracts were written both in Indonesian and English language. A list of the 166 references, with all study characteristics, is presented in Appendix A.

### 3.2. Study Topics

Topics of the included studies comprised research on measurement and prevalence, correlates, and outcomes of physical activity and sedentary behavior as well as a validation study. Of the 166 included studies, physical activity-only studies comprised the largest proportion (67.5%), followed by both physical activity and sedentary behavior studies (17.5%) and sedentary behavior-only studies (15%). In 28% of studies, the primary research focus was not on physical activity, but the prevalence or measurement of physical activity was reported. Similarly, 15.9% of studies were not primarily focused on sedentary behavior but reported the prevalence or measurement of sedentary behavior. Nearly a quarter of the studies investigated correlates of physical activity (see Appendix A). Body mass index (BMI) was the focus of two-thirds of studies (*n* = 50) examining correlates of physical activity and was also the focus of 61% of the studies (*n* = 25) investigating correlates of sedentary behavior. Physical fitness accounted for 66% of studies reporting outcomes of physical activity (see Table 1).

At least 12 nationally representative studies reported the prevalence of physical activity and sedentary behavior (see Table 2). The definition of sufficient physical activity in children and adolescents varied in the included studies, from obtaining at least 150 min of MVPA per week [31,32], doing 60 min of MVPA per day at least five days per week [33,34,35,36], to obtaining at least 60 min of MVPA daily [37,38]. The prevalence ranges between 12.2% and 52.3% for “sufficient” physical activity, and between 24.5% and 33.8% for sedentary behavior ≥3 h per day (see Table 2).

### 3.3. Research Designs Used

Most studies used a quantitative design (98.8%) and the rest were mixed-methods (1.2%). More than half of the studies were cross-sectional (56%), and 30.1% of the studies used an intervention trial design. While the most common sampling methods were purposive (28.3%) and random sampling (22.3%), just over 10% of the studies did not provide clear information on the sampling method (see Appendix A).

### 3.4. Characteristics of the Study Sample

The majority (78.9%) of the studies involved both female and male participants. Sample size ranged from 20 to 1,017,290 participants. There were limitations in identifying sample sizes in some nationally representative studies as they did not specify the sample sizes of each age group.

While 98.2% of studies reported the location of the study, more than 80% did not report its geographical type (i.e., rural or urban area). Just over 60% were conducted on Java Island, and 11.4% of studies were national population-based studies. Nearly one-third of the studies investigated children at the primary school level (±7–12 years old), and nearly 30% examined adolescents at the senior high school level (±16–18 years old) (see Appendix A).

### 3.5. Measurement of Physical Activity and Sedentary Behavior

Of the 141 studies that examined only physical activity or both physical activity and sedentary behavior, 90 studies (63.8%) reported the measurement tools. Of these, 81.1% utilized questionnaires. The only study that utilized an accelerometer was published in 1998 [21]. Other studies that used device-based measurement, all of which used a pedometer, were found in 2013 [42], 2015 [43], and 2018 [44]. A high proportion (86.7%) of the studies measured total physical activity rather than specific domains. Most studies (92.2%) failed to provide information on the validity of the instruments.

Of the 54 studies investigating only sedentary behavior or both physical activity and sedentary behavior, 52 (96.3%) reported the assessment tools. Of these, 88.5% utilized questionnaires. The rest of the studies collected data using a diary (7.7%), interview and observation (1.9% each). There were no sedentary behavior studies that utilized device-based measurement. Screen time and total sedentary time were the focus of 36.5% and 30.8% of the studies, respectively. The majority of the studies (90.4%) did not provide information on the validity of the instruments (see Table 3).

## 4. Discussion

This review aimed to locate and analyze research trends in physical activity and sedentary behavior studies in Indonesian youth (7–18 years old), as well as to map associated research designs, with a view to identify gaps in the literature and to propose directions for future research.

### 4.1. Trends in Physical Activity and Sedentary Behavior Studies

Our results show that the number of physical activity and sedentary behavior studies in Indonesian youth was relatively low. The chaos of political reformation in 1998 [45] may explain the absence of publications in these areas during 1999 and 2000. Nevertheless, there was a significant increase in the number of publications, particularly after 2011. This trend seems consistent with findings from other LMICs [16,46]. The growth of published studies after 2011 was in line with the growth of overall published studies in Indonesia. This increase can be attributed to the development of online and open access journals in Indonesia, which started to exponentiate in 2011 [47]. It has made studies more accessible compared to previous eras when most journals in Indonesia were paper-based. The increase in publications might also be attributed to the series of policies in 2011 and 2012 by the Directorate General of Higher Education, a division in the Indonesian Ministry of Education and Culture [48]. Since the policies were enacted, it has been compulsory for higher education lecturers and students to publish articles in online and reputable journals as one of pre-requirements for promotion or graduation [48]. Lecturers were also encouraged to store their unpublished works in university repositories, which is reflected in the increase in university repository volume in Indonesia [48]. It is however unclear why the studies regarding physical activity decreased after 2017. A lack of clear policy and support for physical activity research may contribute to the problems. The gradual decrease in the number of physical activity studies after 2017, and the limited number on sedentary behavior, signify a need to investigate both topics further in Indonesian youth.

Similar to other LMICs, as well as the wider international literature, the number of physical activity studies was higher than that for sedentary behavior in Indonesian youth [16,46]. In comparison with physical activity epidemiology, sedentary behavior research is much more recent [16], which may provide a reason for the smaller number of sedentary behavior studies. While the landmark study in physical activity epidemiology was published in 1953 by Morris and colleagues [49], the first publication of a physical activity-related study on Indonesian youth was found in 1998 [21]. Meanwhile, the first publication reporting on sedentary behavior in youth was found in 2004 [22]. This finding is consistent with global studies of sedentary behavior that increased sharply in the early 2000s [50] and the literature of sedentary behavior in Bangladesh, another Asian LMIC [46].

### 4.2. Research Designs Used

Some key findings relate to the methodology of the included studies. Compared to the majority of the study designs on physical activity and sedentary behavior research in other Asian LMICs [16,46] and globally [51,52], this review revealed a large proportion of cross-sectional studies. Although a cross-sectional design provides some benefits, including time- and cost-effectiveness, it has clear limitations, including the inability to infer causation. In line with the suggestion from the scoping review study in Thailand [16], more longitudinal and intervention studies are warranted to increase the robustness of conclusions regarding causality and determinants of physical activity and sedentary behavior in Indonesian youth. This is a key finding for the progressive development of knowledge concerning physical activity and sedentary behaviors in Indonesian youth. For example, creating a robust policy in Indonesia will require a level of evidence higher than mere cross-sectional designs.

The lack of qualitative and mixed-methods studies requires more attention. If conducted appropriately, qualitative methods allow for a deep, nuanced, and multi-layered understanding and interpretation of thoughts and behaviors [53]. Meanwhile, mixed-methods studies—those using both quantitative and qualitative methods—are recommended in health and behavior change research [54]. This study design offers the ability to derive a more comprehensive understanding of the research issues by integrating information from both quantitative and qualitative approaches. This process may counterbalance the strengths and weaknesses of each method [55].

Similar to the majority of physical activity and sedentary behavior research in LMICs [16,46] and globally [51], self-report questionnaires were widely used to assess participation and prevalence. However, the validity of almost all questionnaires in the included studies is unclear. Validation and cross-cultural adaptation studies of the best available international questionnaires are now warranted and previous scoping review studies in LMICs also recommend this [16,46]. Moreover, while self-reported methods have their weaknesses, some domains of both physical activity and sedentary behavior are best assessed this way. For example, screen time rather than total sedentary time is often associated with poor health outcomes in young people. Assessments using only devices will not necessarily capture this at all or in the detail needed.

Nevertheless, it is essential to note that measuring physical activity and sedentary behavior by using only questionnaires can influence data quality, with known limitations, including recall and social desirability biases [54]. In line with the suggestion from the scoping review studies in LMICs [16,46], it is encouraged for future studies to use device-based measurements more. Device-based measurements can provide more valid and reliable data, particularly for total time spent in different intensities of movement, as well as temporal patterning across the day or week. However, the affordability of such devices is problematic and may explain the lack of usage in Indonesian research. Taking part in internationally funded projects and collaborating with international universities may become a feasible option, as well as allowing the sharing of devices. If appropriate for the research project, pedometers could be used, and these will be significantly more affordable. However, these will limit the researcher to assessing only the domain of ambulation. Partnerships with commercial companies may be another way to access devices in a cost-effective and sustainable way.

Aligned with previous studies in LMICs [16,46], this scoping review revealed that the majority of the included studies have a limited sample size. The majority of included literature was centered on Java—the island where the capital city and the central government offices of Indonesia are located. With a geographical area of 7% of Indonesia, Java is inhabited by 57% of the total population [44]. Nearly half of the universities—as a common base of the researcher—are located in Java [45], which may explain the higher number of physical activity and sedentary behavior studies compared to other islands. Due to the uniqueness of the geographical situation in Indonesia, which is spread across numerous islands, physical activity and sedentary behavior researchers may experience challenges in researching multiple islands. To overcome this issue, researchers may need to conduct a collaborative study with researchers from other regions or countries, and other related fields, to share the costs and expand the scope of the research, including conducting research with a larger sample size and with nationally representative samples. This could provide a broader understanding of physical activity and sedentary behavior in Indonesia.

Another finding related to the methods located in this scoping review is the lack of information concerning the methodology of the studies, such as data collection methods, measurements, and validity of the instruments. With the expansion in the number of standardized guidelines for reporting different types of research (e.g., CONSORT-social and psychological interventions (SPI) 2018 [56]), researchers in the physical activity and sedentary behavior field in Indonesia should be made aware of such protocols and encouraged to make greater use of them.

A final finding is that the majority of the included studies investigated young people at the primary school (±7–12 years old) and the senior high school (±16–18 years old) level. Future studies may focus more on youth at the junior high school level (±13–15 years old). Research in this age group may provide interesting insights as this period is a transition period from childhood to young adulthood, where young people have a greater degree of freedom to do and choose activities than when they were in younger ages.

### 4.3. Study Topics

Physical activity and sedentary behavior studies in Indonesian youth were mostly reporting on prevalence/measurement, correlates, and a limited number of outcomes of physical activity. A significant proportion of correlates of physical activity studies focused on BMI, and physical fitness was the most often studied outcome of physical activity. Future studies need to expand the focus to other correlates of physical activity and explore the correlates of sedentary behavior as well.

As Bauman et al. point out, few studies in LMICs investigate the association between physical activity and psychological, cognitive, affective, social, and cultural factors [57], and future studies need to address this. Future studies in physical activity and sedentary behavior also need to investigate the environmental correlates [16]. A previous study found that walkability, traffic speed/volume, land-use mix (access from home to destinations such as schools and shops), and residential density are among the correlates of physical activity in children and adolescents [58]. However, the majority of the included studies in that review originate from high-income countries, showing the need to check the relevance of the results in LMICs. Studies investigating in-depth personal views of physical activity and sedentary behavior in youth are also warranted to reveal nuanced reasons behind the physical activity and sedentary behavior level of each individual.

The prevalence of “sufficient” physical activity ranges between 12.2% and 52.3%, while the prevalence of sedentary behavior ≥3 h per day ranges between 24.5% and 33.8% in Indonesian youth. This is similar to results among Southeast Asian countries, LMICs, and globally [33,34,35,59]. In a study involving data from 105 countries, Hallal et al. found that around 20% of adolescents engaged in 60 min or more of MVPA per day [60]. Meanwhile, data from 40 countries in Europe and North America showed that around two-thirds of adolescents spend ≥2 h per day watching television [60]. Aligned with the recommendation from the scoping review in Bangladesh [46], there is an urgent need to promote physical activity and to limit sedentary behavior in a more massive way among Indonesian youth. It is also crucial for future studies to conduct good prevalence studies with robust measures so these can inform intervention studies. Moreover, there is an urgent need to update current policy and to develop a national guideline on physical activity and sedentary behavior based on specific ages in Indonesia.

Up until now, there is no specific national guideline on physical activity and sedentary behavior in Indonesia. While there have been some systems in place for talent scouting (i.e., the National Student Sports Olympiad) and physical education within school systems, physical activity promotion among school children has been very limited. Moreover, although some efforts to promote physical activity at the community level have been initiated at the national level since January 2017 through the “Healthy Lifestyle Community Movement” by the Indonesian president, in which physical activity promotion was identified as one of the key elements [61,62], systematic effort to increase physical activity and sedentary behavior among school children is still scarce. The guidelines for physical activity (frequency, intensity, type, time) and sedentary behavior for Indonesian youth are not yet available [62]. It might be due to limited research in these areas, as highlighted as one of the major findings of this scoping review. More research in these areas is thus recommended to guide the development of policies for promoting physical activity and reducing sedentary behavior, best suited for Indonesian school children.

### 4.4. Strengths and Limitations

The key strength of this review includes the strategy to maximize the results for both published and gray literature by searching platforms both in national and international settings, using a wide range of syntaxes, and capturing literature both in the Indonesian and English language. This review is, however, also subject to some limitations. Firstly, but consistent with the aims and conventions of scoping reviews, we did not assess the quality of the studies. Secondly, we searched only two Indonesian platforms, which may exclude relevant literature from other Indonesian repositories. Nevertheless, we used both Google and Google Scholar platforms to address this issue as these can index literature from universities.

## 5. Conclusions

This scoping review revealed that while there was a significant increase in the number of physical activity and sedentary behavior studies in Indonesian youth, especially after 2011, there was a gradual decrease in the number of studies after 2017, which signifies a need to investigate both topics further in Indonesian youth. This review shows that the high prevalence of insufficient physical activity and high sedentary behavior in Indonesian youth is suggestive of a crucial need to update policy, to develop national guidelines on physical activity and sedentary behavior based on specific ages, and to do more massive promotion to Indonesian youth to increase their physical activity level and to limit sedentary behavior.

The gaps and limitations of previous studies include the large proportion of cross-sectional studies, the lack of qualitative and mixed-methods studies, the excessive use of self-report questionnaires, and the limited sample sizes that were centered on Java island. Other limitations are the lack of information regarding the research methodology, the limited number of studies in youth at the junior high school level (±13–15 years old), and that previous studies mostly reporting on prevalence/measurement, correlates, and a limited number of outcomes of physical activity.

It is recommended for future studies to do more longitudinal, intervention, qualitative and mixed-method studies. Validation and cross-cultural adaptation studies of the best available international questionnaire are also recommended. It is encouraged for future studies to use device-based measurement more, and to conduct research with a larger sample size and with nationally representative samples, e.g., conducting a collaborative study with researchers from other regions or countries, and other related fields. Future studies are also encouraged to use standardized guidelines for reporting different types of research (e.g., CONSORT-SPI 2018 [51]) and to focus more on youth at the junior high school level (±13–15 years old). Regarding the topics, it is recommended for future studies to investigate a wider set of correlates of physical activity and sedentary behavior (i.e., psychological, cognitive, affective, social, cultural, and environmental factors) and to investigate in-depth personal views of physical activity and sedentary behavior.

## Figures and Tables

**Figure 1 ijerph-17-07665-f001:**
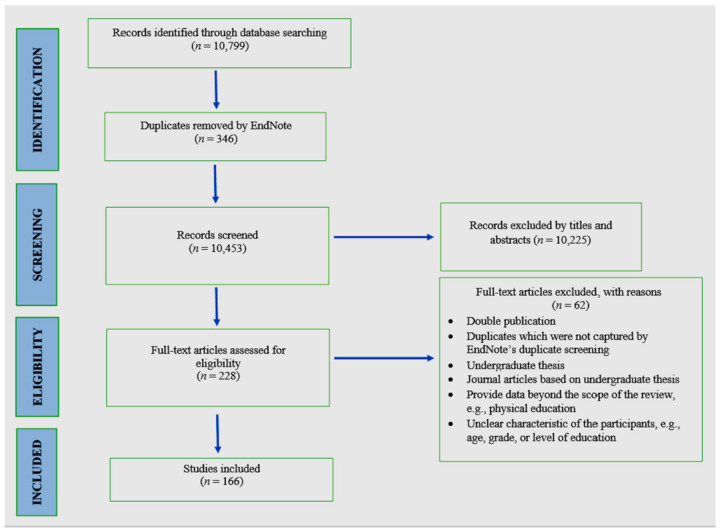
Flow diagram of the study screening process.

**Figure 2 ijerph-17-07665-f002:**
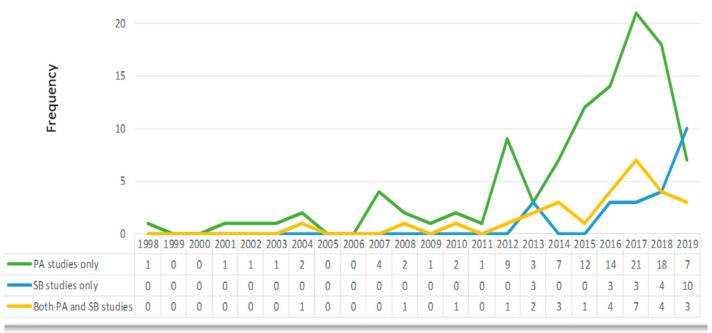
The number of included studies based on the topic, published per year 1998–2019.

**Table 1 ijerph-17-07665-t001:** Number of studies investigating correlates and outcomes of physical activity and sedentary behavior in Indonesian adolescents.

Categories	Correlates of PA		Correlates of SB		Outcomes of PA		Outcomes of SB	
	Number of Studies	%	Number of Studies	%	Number of Studies	%	Number of Studies	%
BMI	50	66.7	25	61.0	6	11.3	-	-
Blood biomarkers	5	6.7	2	4.9	5	9.4	-	-
Physical fitness	6	8.0	-	-	35	66.0	-	-
Socio-demographic	4	5.3	2	4.9	-	-	-	-
Nutritional intake	1	1.3	2	4.9	-	-	-	-
Parental rules	-	-	1	2.4	-	-	-	-
General health	2	2.7	-	-	2	3.8	-	-
Mental health	1	1.3	6	14.6	1	1.9	-	-
Sedentary activity	2	2.7	-	-	-	-	-	-
Motor skill	1	1.3	-	-	2	3.8	-	-
Quality of sleep	1	1.3	2	4.9	-	-	-	-
Memory	-	-	-	-	1	1.9	-	-
Behavior	-	-	-	-	1	1.9	-	-
Drug use	-	-	1	2.4	-	-	-	-
PE Participation	1	1.3	-	-	-	-	-	-
External supports	1	1.3	-	-	-	-	-	-
Posture	-	-	-	-	-	-	1	100.0
Total *	75	100.0	41	100.0	53	100.0	1	100.0

Note: * Multiple correlates and/or outcomes were investigated in some studies; hence, the sum of the totals is greater than the total number of included studies. Number of studies for each topic: Correlates of PA (71 studies), correlates of SB (37 studies), outcomes of PA (49 studies), outcomes of SB (1 study). BMI: body mass index; PE: Physical Education; PA: physical activity; SB: sedentary behavior.

**Table 2 ijerph-17-07665-t002:** Prevalence of physical activity (PA) and sedentary behavior (SB) in Indonesian youth from nationally representative studies.

	Reference	Study Design and Methods	Sample Characteristics	Assessment/Definition of PA or SB	Results
**1.**	Indonesian Department of Health, 2008 [31]	Cross-sectional; interview-administered questionnaire	*n* = 280,000 families; number of samples and age details for age 10–14 years was not reported	Frequency of PA during the past 7 days. Physical inactivity: <150 min MVPA/week (this study used this definition for all age categories).	66.9% samples were inactive, 33.1% did sufficient PA
**2.**	Guthold et al., 2010 [33]	Cross-sectional; self-administered questionnaire	*n* = 2788; age 13–15 years (M = 13.9 years)	The Global School-based Student Health Survey (GSHS) 2007. Sufficient PA: obtaining at least 60 min of PA per day at least 5 days/week. SB: spending 3 or more hours/day on sitting activities.	Prevalence of sufficient PA: 21.5%. Proportion spending ≥3 h SB per day: 33.5%
**3.**	Indonesian Ministry of Health, 2013 [39]	Cross-sectional; self-administered questionnaire	*n* = 1,027,763; number of samples and age details for age 10–14 years not reported	SB was defined as any waking activities characterized by sitting and lying, either in workplace, at home, or during travel.	Prevalence of SB >3 h/day: 71.8%
**4.**	Peltzer and Pengpid, 2016 [34]	Cross-sectional; self-administered questionnaire	*n* = 2867; age 13–15 years	GSHS 2007, definition of sufficient PA and SB is same as reference number 2.	24.4% of participants did sufficient PA. Prevalence of physical inactivity (<5 days/week) = 75.6%, SB (≥3 h) = 33.7%.
**5.**	Arat and Wong, 2017 [35]	Cross-sectional; self-administered questionnaire	*n* = 3116; the average age of male adolescents: 14.85 (SD = 0.91), female adolescents: 14.85 (SD = 0.69)	GSHS 2007, definition of sufficient PA is same as reference number 2.	42.9% of participants did sufficient PA.
**6.**	Permanasari and Aditianti, 2017 [40]	Cross-sectional; questionnaire	*n* = 15,055; age 7–18 years old	Not specified	Prevalence of PA: non-obese group (sufficient: 53.4%, insufficient: 46.6%), obese group (sufficient: 51.2%, insufficient: 48.8%). Overall sufficient PA in both groups: 52.3%.
**7.**	Vancampfort et al., 2019 [9]	Cross-sectional; self-administered questionnaire	*n* = 8806; age 12–15 years old	GSHS 2015. SB: spending 3 or more hours/day on sitting activities.	Prevalence of ≥3 h/day of leisure-time SB: 24.5%.
**8.**	Pengpid and Peltzer, 2019 [37]	Cross-sectional; self-administered questionnaire	*n* = 11,124; mean age 14 years old	GSHS 2015, definition of SB is same as reference number 7. Inadequate PA was defined as not doing at least 60 min of MVPA daily.	Overall, 87.8% of the students had low PA levels (12.2% did sufficient PA). In total, 27.3% engaged in leisure-time SB (≥3 h/day).
**9.**	Vancampfort et al., 2018 [36]	Cross-sectional; self-administered questionnaire	*n* = 3022; mean age: 14 years old	GSHS 2007, definition of sufficient PA and SB is same as reference number 2.	The prevalence of SB ≥3 h/day: 33.8%.
**10.**	World Health Organization, 2018 [38]	Cross-sectional; self-administered questionnaire	*n* = not specified; age 12–15 years old	GSHS 2015. The definition of insufficient PA is same as reference number 8.	Prevalence of insufficient PA among adolescents: 87.1% (12.9 did sufficient PA).
**11.**	Ministry of Health, Republic of Indonesia, 2018 [32]	Cross-sectional; questionnaire	*n* = 1,017,290 for all age categories, details were not specified	Global Physical Activity Questionnaire (GPAQ). Physical inactivity: <150 min MVPA/week (this study used this definition for all age categories)	Prevalence of sufficient PA age 10–14 years old (35.6%), age 15–19 years old (50.4%). Prevalence of insufficient PA age 10–14 (64.4%), age 15–19 (49.6%).
**12.**	Khan, 2019 [41]	Cross-sectional; self-administered questionnaire	*n* = 8731; age 12–15 years old	GSHS 2015 definition of SB is same as reference number 7.	Prevalence of SB (≥3 h/day): 27.3%

**Table 3 ijerph-17-07665-t003:** Instruments for measuring physical activity and sedentary behavior.

Instrument Type	Physical Activity		Sedentary Behavior	
	Frequency	%	Frequency	%
Questionnaire				
Modified Children’s Physical Activity Questionnaire (CPAQ)	1	1.1	3	5.8
International Physical Activity Questionnaire (IPAQ)/Modified IPAQ	7	7.8	1	1.9
Global Physical Activity Questionnaire (GPAQ)	5	5.6	1	1.9
The Activity Participation Questionnaire (APAQ)	1	1.1	-	-
PAQ-A (Physical Activity Questionnaire for Adolescent)	6	6.7	-	-
Modified the General Practice Physical Activity Questionnaire (GPPAQ) and the Physical Activity Level (PAL)	1	1.1	-	-
Adolescent Physical Activity Recall Questionnaires (APARQ)	2	2.2	-	-
Physical Activity Questionnaire for Older Children (PAQ-C)	7	7.8	1	1.9
Global School-based Student Health Survey (GSHS)	6	6.7	11	21.2
The Indonesian Online Game Addiction Questionnaire	0	0.0	2	3.8
ASAQ (Adolescents Activity Sedentary Questionnaire)	0	0.0	7	13.5
The Sedentary Behaviors Questionnaire (SBQ)	0	0.0	1	1.9
Other Questionnaires	10	11.1	8	15.4
Not specified	27	30.0	11	21.2
Subtotal	73	81.1	46	88.5
Interview guideline				
Subtotal	3	3.3	1	1.9
Diary				
Activity record form Diary	4	4.4	3	5.8
Bouchard diary	1	1.1	1	1.9
Trial of Activity for Adolescent Girls (TAAG) Diary	1	1.1	-	-
3 × 24 daily activities diary	1	1.1	-	-
Daily Physical Activity (DPA) Card Diary	1	1.1	-	-
Subtotal	8	8.9	4	7.7
Device-based				
Pedometer	3	3.3	-	-
Accelerometer	1	1.1	-	-
Subtotal	4	4.4	-	-
Test				
Unspecified test	1	1.1	-	-
Subtotal	1	1.1	-	-
Observation				
Observation sheet	1	1.1	1	1.9
Subtotal	1	1.1	1	1.9
Total	90	100.0	52	100.0

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
