# Peer review of "Physical Activity and Sedentary Behavior Research in Indonesian Youth: A Scoping Review"

_ijerph, 2020, doi:10.3390/ijerph17207665_

Round 1
Reviewer 1 Report
Well written manuscript.
My one suggestion to you is defining 'scoping' in your article. On line 60 scoping review is mentioned. Yes, this is a scoping review. It is also a new and up and coming term. I think it would benefit the reader to give a brief definition to scoping review. I think after you give a brief definition, that should help the reader.
166 studies. I like that.
Reviewer 2 Report
This study has carefully reviewed the research on physical activity and sitting behavior in Indonesian youth. However, it is unlikely to be of interest to non-Indonesian readers, and I think the findings described are somewhat lacking in novelty and originality. Therefore, I think it is desirable to submit to an academic journal for readers for Indonesia.
Reviewer 3 Report
Congratulations. Excelent review
Reviewer 4 Report
The authors conducted a scoping review regarding the overall trend in physical activity and sedentary behavior research among Indonesian youth. The review comprehensively covers the wide range of methodological and participant characteristics and yields helpful findings to design future studies in this field. This reviewer made several comments primarily on method and discussion sections as listed below.
- Why was Indonesia chosen as a topic of this scoping review out of LMICs in Asia? The authors mentioned Indonesia’s public health and economical situations from lines 48 to 53. However, it is not sufficiently convincing to this reviewer.
- Line 63: This reviewer suggests that the authors writes “research designs” instead of “research methods” here and throughout the manuscript.
- Line 68: This reviewer recommends that the author team attaches the PRISMA-ScR checklist and shows each location mentioned within the manuscript corresponding to checklist item. This reviewer finds that several checklist items are missing.
- Line 71: Pubmed -> PubMed
- Line 73: Why was the Gadjah Mada University’s database only included? Why not other university’s databases? It feels simply strange to this reviewer.
- Why was SPORTDiscus not included? It is one of the databases specific to sport and health sciences.
- Line 80: Exclusion criteria (1) is not necessary because it is just a mirror of inclusion criteria (1).
- Why did the authors exclude studies regarding physical education? PE must be one of the most common form of physical activity/sedentary-reducing intervention among school-going youth. This may be one reason for smaller number of intervention studies identified through your scoping review.
- Line 129: The authors should provide citations for the definitions of sufficient physical activity.
- Line 133: Was the definition of prolonged sedentary behavior varied? If so, explain this and insert citation(s) for it.
- Line 137: If available, could the authors describe the distribution of the intervention study designs, namely pre-post, non-RCT, and RCT?
- Line 147: If available, please provide ethnicity distribution of the study sample.
- Is there Active Healthy Kids Report Card for Indonesians? If available, cite it and discuss further regarding the relation of your findings with the Report Card.
- Table 2, ref 1, results: 66,9% -> 66.9%
- Line 161: Please double-check this sentence for grammatical correctness.
- Line 180: Please discuss further why a significant increase in the number of literatures. Is there any trigger (e.g., political transition, megaevent, landmark paper, etc) to explain this?
- Dividing the discussion section with several appropriate subheadings would increase readability of your manuscript.
- The authors should overview Indonesia’s current policies related to youth physical activity and sedentary behavior and discuss further how to update them by considering your findings through the scoping review.
Reviewer 5 Report
This scoping review of physical activity and sedentary behaviour research in Indonesian youth makes an important contribution to the existing evidence base, and represents a crucial first step in improving these behaviours in the population. The review was generally very well conducted and written, although before publication I do feel there is a need for further justification and information about some of the procedures.
Introduction
- Please include more specific justification for looking at Indonesian youth as much of the data provided is in adults, and the focus on young people feels like an afterthought. You could think about risk factors for NCDs in young people and how these relate to PA and SB as well as long-term trends in PA and SB- do behaviours in young people track through to adulthood?
- When referring to PA and SB guidelines please state what these are so the reader has a reference point.
- Please justify why only looking at Indonesia. Have there been other similar scoping reviews or systematic reviews in other LMICs? What have these shown? Why might it be important that reviews are country/context specific?
Methods
- Figure 1 indicates that 46 studies were identified through other sources. Please provide details of what these are in text- i.e. was this through screening reference lists, emailing authors etc
- In figure 1 please provide numbers for each of the reasons for inclusion
- Provide justification for your inclusion and exclusion criteria. Specifically, why an age range of 7-18 years? Why were PE specific studies excluded?
- Please include the completed PRISMA scoping review checklist as supplementary material
Results
- I understand why 2020 studies are not included in figure 2 but please highlight in text how many studies there were up until April.
- How was a nationally representative study defined?
- Did you assess study quality at all?
- Given the broad age range it would be good to explore age within the included studies- are they relatively evenly spread across the age range or predominantly in younger/older age groups? This could have implications for what is recommended
Discussion
- You provide several ideas of how device-based measurement of PA and SB could be used in research in LMICs, but can anything be learnt from research in other LMIC's? Are there any examples of good practice here?
- You highlight that a strength of this scoping review is the search strategy in terms of maximising results for both published and grey literature yet your inclusion criteria is focused on just a few types of publications. Can you really say that grey literature are included?
Round 2
Reviewer 2 Report
I think this revised paper has improved somewhat by adding the perspective of analysis of several departments. However, I don't think that new knowledge or perspectives have been proposed other than targeting Indonesian youth. Nothing is particularly bad, but I feel that it lacks the knowledge appropriate for adoption.
Reviewer 4 Report
The authors appropriately addressed all of this reviewer's comments. I do not have any further suggestions.